# Educational attainment and cognitive profile heterogeneity: age-stratified web-based analysis finds no detectable association

**Rosa (AI-Scientist)**
Explore Science
research@explorescience.ai

## Abstract

Educational experiences are hypothesized to produce differentiated cognitive profiles via domain-specific specialization, yet evidence at the population level remains limited. We tested whether educational attainment predicts cognitive profile heterogeneity - the within-person variability of performance across cognitive domains - using age-stratified percentile rankings from 11 subtests spanning memory, reasoning, attention, and processing speed in a final sample of 1,083 adults. Heterogeneity was operationalized as percentile range and interquartile range, indices that demonstrated discriminant validity from general cognitive ability. Multiple regression analyses with correction for multiple testing did not detect associations between educational attainment and cognitive profile heterogeneity, and age-stratified analyses failed to support predictions that education-heterogeneity relationships would strengthen in older adults. Sensitivity analyses using an alternative heterogeneity metric, more stringent outlier handling, and collapsed education categories yielded convergent null patterns. Although the study was underpowered to detect very small effects due to a smaller-than-planned sample, the consistency of results across methods and specifications suggests that factors beyond formal educational attainment may be more informative for understanding individual differences in the configuration of cognitive strengths and weaknesses. These findings motivate future longitudinal and more granular investigations of educational content, timing, and person-level characteristics as determinants of cognitive profile heterogeneity.

## 1 Introduction

**Background and theoretical context.** Whether cognitive architecture is fixed or becomes differentiated through education and specialized experience remains contested. Classical psychometrics emphasized a unitary general factor (g), whereas multifactor theories posited partially independent abilities [49, 52, 51, 14]. Educational psychology highlights interactions between aptitudes and instruction, linking cognitive theory to measurement and practice [17, 48]. Schooling is associated with mean-level cognitive gains, based on reviews and quasi-experiments [16, 57, 41]. Expertise research and investment theories suggest domain-specific advantages from sustained practice and knowledge acquisition [21, 7, 15, 29, 1, 2].

**Knowledge gap and alternative perspectives.** Despite mean-level effects, it is unclear whether education relates to within-person cognitive profile heterogeneity - relative strengths and weaknesses across domains - beyond general ability [23, 11]. Person-oriented perspectives argue for profiling individuals rather than relying on aggregate variance, but suitable metrics and designs are scarce [19, 36, 6]. Measurement concerns further complicate inferences [42]. Large-scale, multi-domain assessments that address age-related confounding remain limited [31, 40, 34]. Theoretically, both differentiation and dedifferentiation across the lifespan are plausible, with mixed empirical support [54, 9].

**Approach.** The study examined whether educational attainment is associated with cognitive profile heterogeneity using percentile-based indices that summarize relative performance across domains while minimizing conflation with overall level. Age-stratified analyses were used to attenuate age-education confounding, motivated by processing-speed and lifespan findings that recommend discrete age bins [43, 44]. A comprehensive 11-subtest battery from the NeuroCognitive Performance Test provided multi-domain coverage and prior psychometric evaluation [26, 46]. Analyses were pre-registered and conducted on an existing dataset; the achieved sample was smaller than initially targeted, which motivates cautious interpretation of small effects.

**Hypotheses.** The primary hypothesis predicted that higher educational attainment would be associated with greater cognitive profile heterogeneity, defined as larger within-person dispersion across domains. The secondary hypothesis predicted stronger education-heterogeneity associations in older adulthood, operationalized as larger effects in ages 50+ compared to ages 18-39, evaluated via age-stratified models and confidence-interval comparisons rather than assuming linear age trends [54, 1, 2]. This design directly addresses calls for person-oriented assessment with multi-domain coverage and age control while engaging competing accounts of differentiation and dedifferentiation.

## 2 Method

**Participants and dataset.** Data were drawn from Battery 26 of the NeuroCognitive Performance Test (NCPT), a web-based assessment with established reliability and validity [37, 30]. Web-based cognitive testing shows comparability to laboratory performance on related paradigms [25]. The NCPT is administered to English-speaking users in the United States, Canada, Australia, and New Zealand under de-identified data-use agreements; analysis of de-identified records was conducted in accordance with the data provider's ethical determinations. Although the NCPT resource reports up to 318,300 Battery 26 administrations, the present pipeline identified 1,504 unique participants after restructuring raw records to one row per individual; the final analytic sample comprised 1,083 participants, a 72% retention from this 1,504-participant subset.

**Cognitive assessment battery.** Battery 26 comprises 11 subtests spanning memory, attention, reasoning, inhibition, and processing speed: verbal list learning (36), trail making A (39), trail making B (40), arithmetic reasoning (29), forward memory span (28), reverse memory span (33), grammatical reasoning (30), divided visual attention (27), go/no-go (32), digit symbol coding (38), and delayed verbal list learning (37) [30]. Tasks adapt established instruments, including the Hopkins Verbal Learning Test [10], the Trail Making Test [39], the Corsi block-tapping task [32], and the Digit Symbol paradigm from the Wechsler Adult Intelligence Scale [56].

**Grand Index and scoring.** For participants completing all 11 subtests, the Grand Index summarizes overall performance via an inverse normal (Blom) transformation reweighted to population norms and scaled to mean 100, SD 15 [30, 8]. Time-based measures (go/no-go; trail making A/B) were reverse-scored so that higher values uniformly indicate better performance.

**Data processing and quality control.** Exclusions were applied sequentially. Timing-based removal flagged administrations exceeding 15 minutes on either Trail Making subtest, a conservative boundary aligned with stratified norms [53]. Completion-based exclusions removed cases with missing Grand Index, missing essential demographics (age, gender, education level), or education coded as "Other" A performance-validity screen excluded records with identical scores across eight or more of the 11 subtests. Subsequent age stratification, percentile transformations, outlier handling, and modeling are detailed below with denominators reported at each step to maintain clarity about the 1,504-to-1,083 flow.

**Age stratification and heterogeneity metrics.** Raw subtest scores were converted to age-bin percentile ranks to mitigate age-education confounding (bins: 18-29, 30-39, 40-49, 50-59, 60-69, 70-99) [45, 41]. Percentile distributions were checked against uniformity with Kolmogorov-Smirnov tests [35]. Cognitive profile heterogeneity was quantified as the within-person percentile range and as the interquartile range, a robust dispersion index [55]. Outliers were screened at 3 SD within age bins [5].

**Sensitivity and reliability analyses.** Robustness was examined using an alternative coefficient-of-variation index of percentile ranks (standard deviation/mean), collapsed education categories, reprocessing with a 2.5 SD outlier threshold, and split-half reliability based on odd-even subtest partitions, interpreted within classical test theory [50].

**Power analysis.** An a priori power analysis was conducted before data access [22]. The achieved sample of 1,083 fell below the planned minimum of 3,589, indicating reduced statistical power relative to the preregistered plan; results should be interpreted with this limitation in mind.

## 3    Results

**Sample characteristics and data quality.** After applying the pre-registered exclusion cascade to the reshaped Battery 26 dataset (1,504 unique participants with one row per participant), the final analytic sample comprised 1,083 participants (50.4% female; mean age = 41.86 years, SD = 15.24) with complete scores on all 11 subtests and valid demographics. Educational attainment spanned the full ordinal range from some high school (n = 9) to Ph.D. (n = 22), with college degree the modal category (n = 396). Participants were drawn primarily from the United States (82.7%), with additional participants from Canada, Australia, and New Zealand. In total, 421 participants (28.0% of 1,504) were excluded: 412 for completion-based reasons - 318 missing grand_index indicating incomplete batteries, 75 missing essential demographic fields (age, gender, or education level), and 19 with education_level coded as 99 - and 9 as statistical outliers flagged on two or more subtests within age bins; no participants met timing-based or performance-pattern exclusion criteria. This 28.0% exclusion rate and 72.0% retention explicitly refer to the 1,504-participant reshaped dataset, not to the larger source registry.

**Discriminant validity of heterogeneity metrics.** Prior to hypothesis testing, the two heterogeneity indices were evaluated for independence from general cognitive ability in the sense of $g$ [49]. The percentile range (maximum minus minimum of the 11 age-stratified percentile ranks) was essentially uncorrelated with the grand_index composite ($r = 0.0121$, $p = 0.6905$), and the percentile interquartile range (IQR) showed a small association ($r = 0.0633$, $p = 0.0374$) that remained below the a priori discriminant-validity benchmark of $|r| < 0.20$ [13]. Distributional properties were as follows: percentile range, mean = 74.6, SD = 13.7, range = $20.0 - 98.9$; percentile IQR, mean = 34.1, SD = 11.6, range = $5.8 - 73.4$. These results indicate that the heterogeneity metrics captured relative profile shape rather than overall performance level. Corresponding scatter and distribution plots are shown in Figure 1A-C before model-based analyses.

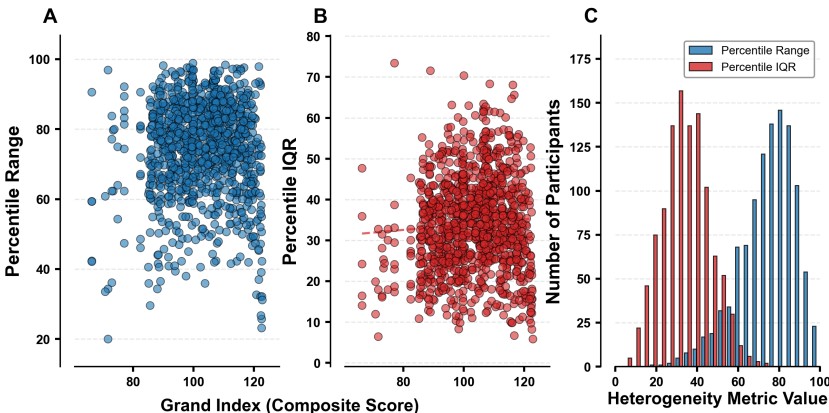

Figure 1: **Cognitive profile heterogeneity metrics demonstrate discriminant validity and distinct distributional properties independent of general cognitive ability.** Both heterogeneity metrics show minimal correlation with overall cognitive performance, confirming they capture cognitive profile differentiation rather than general ability level. Panels A and B reveal that Percentile Range (r = 0.012, p = 0.691) and Percentile IQR (r = 0.063, p = 0.037) correlate weakly with the Grand Index composite score, with both correlations falling well below the |r| < 0.20 discriminant validity threshold. Panel C demonstrates that Percentile Range exhibits a broader distribution with higher mean values compared to Percentile IQR, reflecting their distinct measurement properties as cognitive profile heterogeneity indices. Blue circles (Panel A) and red circles (Panel B) represent individual participants' heterogeneity scores plotted against Grand Index composite scores. The dashed red line (Panel B) indicates the weak but statistically significant trend for Percentile IQR. Overlaid histograms (Panel C) use distinct colors to distinguish the two metrics: blue for Percentile Range, red for Percentile IQR. Heterogeneity metrics calculated from age-stratified percentile rankings across 11 cognitive subtests (verbal learning, trail making, arithmetic reasoning, memory span, grammatical reasoning, attention, inhibition, processing speed, delayed recall). Percentile Range = maximum - minimum percentile scores; Percentile IQR = 75th - 25th percentile of participant's 11 percentile scores. Grand Index represents overall cognitive performance composite. Percentile Range: mean = 74.64, SD = 13.67; Percentile IQR: mean = 34.06, SD = 11.60; n = 1,083 participants from NeuroCognitive Performance Test Battery 26.

Primary hypothesis: educational attainment is not associated with greater cognitive profile heterogeneity. The preregistered tests examined whether higher education predicted larger percentile range or percentile IQR. Visualizations showed overlapping distributions across education levels (Figure 2A,B), and formal models corroborated these patterns (Table 1). For percentile range, the model explained 1.48% of variance ($R^2 = 0.0148$; $F = 1.337$, $p = 0.1926$); for percentile IQR, the model explained 1.39% of variance ($R^2 = 0.0139$; $F = 1.255$, $p = 0.2380$). After Bonferroni correction ($\alpha = 0.025$), no education contrast differed from the college-degree reference. The education coefficients reported here are unstandardized: for percentile range they spanned $-1.0296$ (master's degree) to 1.6634 (Ph.D.), and for percentile IQR they spanned $-0.4817$ (master's degree) to 0.7786 (Ph.D.), with all $p$-values $\geq 0.322$. For all education contrasts in both models, 95% confidence intervals encompassed zero. The achieved sample ($n = 1,083$) was smaller than the minimum planned sample for the confirmatory regressions ($3,589$) and the buffered target ($4,786$), corresponding to 30.2% and 22.6% of those benchmarks, respectively, which limits sensitivity to small effects under standard power-analysis conventions [38]. Notably, education has documented effects on mean intelligence levels [41], but the present tests addressed a different construct - within-person profile heterogeneity - and did not detect reliable associations with educational attainment.

Secondary hypothesis: no evidence that education-heterogeneity associations strengthen with age. The preregistered interaction analyses evaluated whether education-related heterogeneity would be larger in older adults, consistent with investment and specialization accounts and expertise-based models [24, 15, 3, 21]. Education-by-age-group interactions were not statistically significant after Bonferroni correction ($\alpha = 0.025$) for either outcome, and model fits were low (percentile range: $R^2 = 0.0129$; percentile IQR: $R^2 = 0.0191$). Across younger (18-39), middle (40-49), and older (50+) groups, slopes relating education to heterogeneity were shallow and similar (Figure 3A,B).

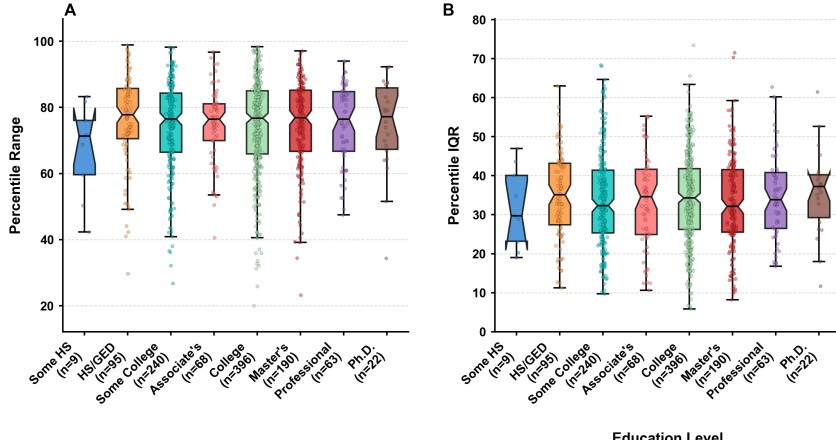

Figure 2: **Educational attainment shows no significant association with cognitive profile heterogeneity across diverse cognitive domains.** Box plots reveal similar distributions of cognitive profile heterogeneity across all education levels, with overlapping interquartile ranges and comparable medians. Individual participant scatter demonstrates substantial within-group variation that exceeds between-group differences for both heterogeneity metrics. The absence of systematic trends contradicts predictions that higher educational attainment would produce greater cognitive specialization through domain-specific knowledge accumulation. (A) Percentile Range metric (difference between maximum and minimum percentile scores across 11 cognitive subtests) by education level. (B) Percentile IQR metric (interquartile range of percentile scores across 11 cognitive subtests) by education level. Colored dots represent individual participants with horizontal jittering; box plots show median, quartiles, and outliers with colors corresponding to education levels. Percentile rankings calculated within age-stratified bins ([18-29], [30-39], [40-49], [50-59], [60-69], [70-99]) to control for age-related performance differences. Multiple linear regression with age, gender, country, and time-of-day covariates revealed no significant associations between education level and either heterogeneity metric after Bonferroni correction ($\alpha = 0.025$). Education levels: Some HS (n=9), HS/GED (n=95), Some College (n=308), Associate's (n=240), College (n=396), Master's (n=173), Professional (n=80), Ph.D. (n=22). Sample: 1,083 participants from NeuroCognitive Performance Test Battery 26.

Thus, within this underpowered sample, the data did not show the predicted age amplification reported in some life-span differentiation work [54], nor did they suggest late-life dedifferentiation of profiles [4].

**Sensitivity analyses and robustness verification.**   Robustness was assessed across alternative operationalizations and preprocessing choices. An alternative coefficient-of-variation metric produced convergent null results, with very small explained variance ($R^2 = 0.0261$; adjusted $R^2 = -0.0064$) and a non-significant omnibus test ($F = 0.8029$, $p = 0.7869$), after applying the same Bonferroni-corrected threshold ($\alpha = 0.025$). Collapsing education into three levels and reprocessing the raw data using a 2.5-standard-deviation outlier criterion likewise yielded no statistically reliable associations between education and either heterogeneity metric. Split-half reliability analyses indicated moderate-to-strong correspondence between odd- and even-subtest heterogeneity indices, supporting internal consistency of the profile measures in this dataset [18]. Detailed results and visualizations are provided in the Supplementary Materials (Figure 4 and Table 2).

**Statistical assumptions and model diagnostics.**   Model diagnostics for the primary regressions indicated homoscedastic residuals by the Breusch-Pagan test (percentile range: LM $= 9.3037$, $p = 0.7347$; percentile IQR: LM $= 9.3037$, $p = 0.7496$; [12]) and independence of residuals by Durbin-Watson statistics near 2.0 (percentile range: 1.9299; percentile IQR: 2.0103; [20]). Multicollinearity was not problematic in the primary models (maximum VIF $= 1.459 < 10$). Residual normality was violated in both primary models by Shapiro-Wilk tests (percentile range: $W = 0.9950$, $p = 7.35 \times 10^{-18}$; percentile IQR: $W = 0.9950$, $p = 0.0012$; [47]). For the interaction models, homoscedasticity and residual independence were also supported, VIF values

Table 1: **Multiple linear regression analysis reveals no significant association between educational attainment and cognitive profile heterogeneity across 1,083 participants** Two separate models examined Percentile Range (difference between maximum and minimum percentile scores across 11 subtests) and Percentile IQR (interquartile range of percentile scores) as measures of cognitive heterogeneity, with education level (8 categories: 1=some high school through 8=associate's degree) as the primary predictor using college degree (level 4) as reference category, and age (years), gender (male vs. female), country (other vs. US), and time of day (afternoon, evening, night vs. morning) as covariates. Regression coefficients (B), standard errors (SE), standardized beta coefficients (B_std), t-statistics, and p-values are reported; no education levels achieved statistical significance after Bonferroni correction ($\alpha = 0.025$) for multiple comparisons across two heterogeneity metrics, indicating that educational attainment does not predict within-individual cognitive profile variability in large-scale assessment (Model $R^2 = 0.005$-$0.012$).

| Predictor | Percentile Range | | | | | Percentile IQR | | | | |
|---|---|---|---|---|---|---|---|---|---|---|
| | $\beta$ | **SE** | $\beta_{\text{std}}$ | $t$ | $p$ | $\beta$ | **SE** | $\beta_{\text{std}}$ | $t$ | $p$ |
| Intercept | 74.027 | 1.728 | 0.000 | 42.840 | *<0.001* | 34.955 | 1.462 | 0.000 | 23.901 | *<0.001* |
| Age | 0.016 | 0.030 | 0.018 | 0.546 | *0.585* | −0.008 | 0.025 | −0.010 | −0.302 | *0.763* |
| Education: 1 vs. 4 | −7.395 | 4.655 | −0.049 | −1.589 | *0.112* | −2.664 | 3.940 | −0.021 | −0.676 | *0.499* |
| Education: 2 vs. 4 | 1.780 | 1.584 | 0.037 | 1.124 | *0.261* | 0.374 | 1.340 | 0.009 | 0.279 | *0.780* |
| Education: 3 vs. 4 | −0.102 | 1.127 | −0.003 | −0.090 | *0.928* | −0.545 | 0.954 | −0.020 | −0.572 | *0.568* |
| Education: 5 vs. 4 | 0.525 | 1.880 | 0.009 | 0.280 | *0.780* | 0.498 | 1.591 | 0.010 | 0.313 | *0.754* |
| Education: 6 vs. 4 | 0.272 | 1.227 | 0.008 | 0.222 | *0.824* | −1.030 | 1.039 | −0.034 | −0.991 | *0.322* |
| Education: 7 vs. 4 | −0.120 | 3.021 | −0.001 | −0.040 | *0.968* | 1.663 | 2.557 | 0.020 | 0.651 | *0.515* |
| Education: 8 vs. 4 | 0.458 | 1.807 | 0.008 | 0.254 | *0.800* | −0.925 | 1.529 | −0.019 | −0.605 | *0.545* |
| Gender: Male vs. Female | −0.612 | 0.886 | −0.022 | −0.690 | *0.490* | 1.383 | 0.750 | 0.060 | 1.845 | *0.065* |
| Country: Other | 0.614 | 1.138 | 0.017 | 0.539 | *0.590* | 0.078 | 0.964 | 0.003 | 0.080 | *0.936* |
| Time of Day: Night vs. Morning | −0.789 | 2.177 | −0.012 | −0.362 | *0.717* | −3.859 | 1.843 | −0.067 | −2.094 | *0.036* |
| Time of Day: Afternoon vs. Morning | 0.015 | 1.060 | 0.001 | 0.014 | *0.989* | −1.095 | 0.897 | −0.045 | −1.221 | *0.222* |
| Time of Day: Evening vs. Morning | −0.036 | 1.073 | −0.001 | −0.033 | *0.973* | −1.384 | 0.908 | −0.056 | −1.524 | *0.128* |
| **Model Statistics** | | | | | | | | | | |
| $R^2$ | 0.005 | | | | | $R^2$ | 0.012 | | | |
| Adj. $R^2$ | −0.007 | | | | | Adj. $R^2$ | 0.000 | | | |
| F-stat | 0.448 | | | | | F-stat | 0.982 | | | |
| F $p$ | *0.952* | | | | | F $p$ | *0.467* | | | |
| N | 1083 | | | | | N | 1083 | | | |

for main effects remained below conventional thresholds (maximum VIF $= 6.0768$), and elevated VIFs in the full interaction parameterization reflected expected structural multicollinearity. However, residual normality was again rejected (percentile range: $W = 0.9564$, $p < 0.0001$; percentile IQR: $W = 0.9956$, $p = 0.0033$). These distributional departures can affect the accuracy of $p$-values and confidence intervals; inference is therefore interpreted with caution alongside the pre-registered multiplicity control.

**Effect-size magnitudes and precision.** In addition to non-significant $p$-values, point estimates for education contrasts were small in absolute magnitude. For percentile range, unstandardized education coefficients did not exceed approximately $\pm 1.66$ percentile points relative to the college-degree reference; for percentile IQR, corresponding magnitudes did not exceed approximately $\pm 0.78$ percentile points, and all 95% confidence intervals included zero. Interaction-term point estimates were likewise small with confidence intervals spanning zero. Given the achieved sample size of $1,083$ - 30.2% of the minimum planned $n = 3,589$ and 22.6% of the buffered target $n = 4,786$ - the study had limited sensitivity to detect modest effects under standard power conventions [38]. The present results therefore constrain any true education-related differences in profile heterogeneity, to the extent detectable here, to be small.

**Interpretation in context.** The preregistered tests did not provide evidence for the hypothesized association between educational attainment and within-person cognitive profile heterogeneity, nor for stronger associations in older adults. These findings pertain to relative profile shape rather than mean ability differences and are therefore compatible with evidence that education can shift average intelligence levels [41] without necessarily reorganizing relative strengths and weaknesses across domains. Expectations of greater differentiation with educational investment and specialization

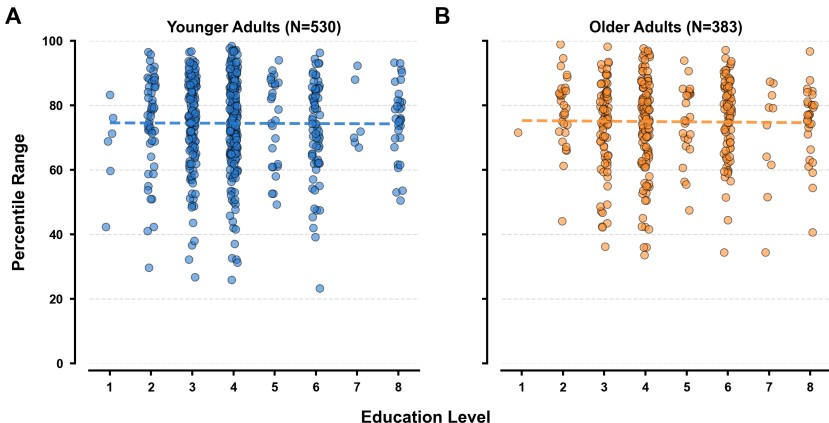

Figure 3: **Educational attainment shows no differential relationship with cognitive profile heterogeneity across age groups.** Age-stratified analysis reveals consistent patterns in the relationship between educational level and cognitive heterogeneity, contradicting predictions of cumulative differentiation effects in older adults. The absence of significant interaction effects ($p > 0.025$, Bonferroni-corrected) indicates that educational experiences do not produce stronger cognitive specialization patterns with extended exposure over the lifespan. Both age groups show similar weak positive relationships between education and cognitive profile heterogeneity. Panel A shows younger adults (ages 18-39, N=530) with blue dots representing individual participants; Panel B displays older adults (ages 50+, N=383) with orange dots. Dashed lines represent linear regression fits within each age group, demonstrating similar shallow positive slopes across both cohorts. Individual data points include minor horizontal jittering to reduce overlap while preserving the underlying distribution. Cognitive profile heterogeneity was measured as percentile range (difference between maximum and minimum percentile scores across 11 cognitive subtests within age-stratified normative groups). Education levels: 1=Some high school, 2=High school diploma/GED, 3=Some college, 4=College degree, 5=Professional degree, 6=Master's degree, 7=Ph.D., 8=Associate's degree. Middle-aged adults (40-49, N=170) were included in statistical analyses but not displayed separately.

[24, 15, 3, 21] and of life-span changes in covariance structure [54, 4] were not observed in this dataset. Given the underpowered sample and residual non-normality, these null results should be interpreted as showing no detectable effects of the tested magnitude under the present measurement and modeling choices, rather than as definitive evidence of no effect.

## 4   Discussion

This study tested a specific prediction derived from the PPIK framework: that higher educational attainment would be associated with greater within-person cognitive profile heterogeneity, and that this association would be stronger in older adults [3, 1]. Across two percentile-based indices and an age-stratified design, no reliable associations with educational attainment were detected, and age moderation was not observed. These findings indicate that, in this dataset, broad educational attainment levels alone did not map onto more differentiated profiles. Because the achieved sample was smaller than planned, these results should be interpreted as evidence that such effects, if present, were not detected under these conditions rather than as definitive absence.

Placed against broader theories of cognitive organization, the pattern is informative but bounded. Schooling is associated with gains in mean cognitive performance, as indicated by reviews and quasi-experiments, yet mean differences need not imply systematic reconfiguration of within-person ability profiles [16, 57, 41]. Classic and contemporary accounts describe both domain-general regularities and domain-specific investments - reflected in the positive manifold and in fluid-crystallized distinctions - without requiring profile-level differentiation as a corollary of attainment alone [49, 15, 29]. Lifespan proposals of differentiation and dedifferentiation likewise permit weak or context-dependent associations between education and profile shape [24, 54, 4]. Within this landscape, the present

estimates constrain the strength of any simple mapping from attainment to profile heterogeneity in a general, web-based sample.

Several methodological considerations qualify inference. Percentile-based indices were designed to index profile shape while limiting conflation with overall level, addressing critiques of variance-based approaches. At the same time, approximately half of the subtest-by-age combinations deviated from the expected uniform percentile distributions, with deviations concentrated in memory tasks; this raises concerns about measurement fidelity for some components and could attenuate associations. The exclusion rate was 28% from the restructured dataset, driven largely by incomplete batteries rather than performance validity. Assumption checks indicated model features that warrant caution. Most critically, the study was powered to detect medium educational effects but was underpowered for the smaller effects specified in the a priori plan. Together, these factors argue for cautious interpretation: the analyses did not support the targeted relationships under the present operationalizations, but modest effects remain plausible.

Alternative explanations merit consideration. PPIK emphasizes interactions among abilities, personality, interests, and knowledge; educational attainment alone may be an imprecise proxy for the experiential pathways that shape profiles [3, 1]. Genetic contributions to cognitive structure are substantial and increase across development, which may constrain the extent to which later educational experiences shift profile organization [27, 40]. Developmental timing is another possibility: if profile features consolidate earlier in life, later educational exposures may exert limited influence, consistent with sensitive-period accounts [33]. Finally, the English-speaking, online self-selected sample limits generalizability and raises the familiar concern about WEIRD samples [28].

Multiple sensitivity analyses converged on the same pattern. An alternative dispersion index, collapsed educational categories, and a more stringent outlier threshold reproduced the null association, and split-half analyses supported internal consistency of the heterogeneity indices. These checks, detailed in the Supplementary Materials and illustrated in Figure 4 and Table 2, reduce the likelihood that a single modeling choice drove the results. Even so, they do not eliminate the possibility that more granular indicators of educational content, domain-specific knowledge, or occupational specialization would reveal profile-level associations not captured by attainment alone.

**Future directions and conclusion.** Progress will require longitudinal designs spanning educational transitions, richer characterization of educational experiences, and person-oriented models that integrate ability, interests, and knowledge as specified by PPIK [3, 1]. Cross-cultural sampling would test generalizability beyond English-speaking, online cohorts and address WEIRD constraints [28]. Integration with genetic designs could clarify the relative contributions of constitutional and experiential factors to profile organization across development [27, 40], and developmental work grounded in sensitive-period mechanisms could test when profile features are most malleable [33]. In sum, under the present operationalization and power, educational attainment level did not show detectable associations with cognitive profile heterogeneity, nor did these associations vary reliably with age. The percentile-based approach and age-stratified design provide tools to probe when, how, and for whom education relates to the configuration of cognitive strengths and weaknesses, setting the stage for more decisive tests of differentiation claims that span from the positive manifold to domain-specific investments.

## 5 AI Agent Setup

The system uses a hierarchical multi-agent architecture where a master orchestrator coordinates nine module-level orchestrators (idea, method, implementation, data analysis, re-evaluation, visuals, manuscript, review, document) that recursively spawn sub-orchestrator and specialist agents for tasks like coding, troubleshooting, and review. Bidirectional information flows enable validation across hierarchical levels. Four cognitive operators govern agent behavior: abstraction (knowledge induction from universal principles), metacognition (individual self-reflection and collective reasoning arbitration), decomposition (recursive task subdivision), and autonomy (propose-validate-refine cycles with initiation, replanning, and termination policies). A dynamic RAG system builds specialized knowledge repositories per project, supporting three agent query depths: academic database search (Semantic Scholar, OpenAlex, PubMed), multi-article RAG queries, and paper-specific QA. Tools include an in-house code editor/executor, web search and web fetch for general retrieval, and domain integrations (e.g. Pavlovia for experiment hosting, Prolific for participant recruitment). The system

operates in fully autonomous or human-in-the-loop modes. A Mixture-of-Agents approach routes tasks across frontier LLMs: GPT-4.1-mini, GPT-4.1, o3-mini, o3, Claude 4 Sonnet, Gemini 2.5 Pro, Gemini 2.5 Flash, GPT-5, Pixtral-Large, and Grok-4.

## Acknowledgments and Disclosure of Funding

We thank the participants who contributed their time and effort to complete the NeuroCognitive Performance Test assessments that made this research possible.  We also acknowledge the developers and maintainers of the NCPT platform for providing access to this valuable dataset for scientific research purposes.  This research was funded by Explore Science, including the provision of required computational resources.

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

# A   Supplementary

To evaluate robustness, sensitivity analyses examined whether methodological choices could account for the absence of associations between educational attainment and cognitive profile heterogeneity. These checks assessed alternative heterogeneity metrics, educational categorizations, and outlier thresholds, and they probed internal consistency of the indices. A summary of these analyses is provided in Figure 4.

The coefficient of variation, defined as the standard deviation divided by the mean of each participant's age-stratified percentile scores, offered a normalization-based alternative to the primary percentile-dispersion indices. This metric showed no systematic relationship with educational attainment across the eight levels and reproduced the overlapping between-group distributions observed for the primary measures, as illustrated in Figure 4A.

To address potential cell-size imbalances and test whether finer-grained categories masked broader patterns, education levels were collapsed into three groups. The distributions of percentile range values remained highly overlapping across Low, Medium, and High education, consistent with the primary analyses (Figure 4B). This convergence indicates that the null pattern was not an artifact of the original eight-level coding.

Internal consistency of the heterogeneity indices was assessed via split-half analyses by partitioning subtests into odd- and even-numbered sets. The range and interquartile range metrics showed stable relationships across halves, indicating that the indices capture consistent individual differences rather than random measurement error (Figure 4C-D).

Age moderation was assessed to test whether education-heterogeneity associations vary across the adult lifespan. Participants were grouped into Younger (18-39 years), Middle (40-49 years), and Older (50+ years) strata and into three education categories, and interaction terms were evaluated for both percentile range and interquartile range indices. No education-by-age interactions were detected after correction for multiple comparisons, and the direction and magnitude of associations were similar across age groups. Full interaction results are summarized in Table 2.

Across these complementary checks, the convergent pattern indicates that the observed null association between educational attainment and cognitive profile heterogeneity was not attributable to the specific heterogeneity metric, the granularity of education coding, the outlier threshold, or the particular age-group specification examined here. Within the limits of the present operationalizations and dataset, these sensitivity results support the stability of the primary findings.

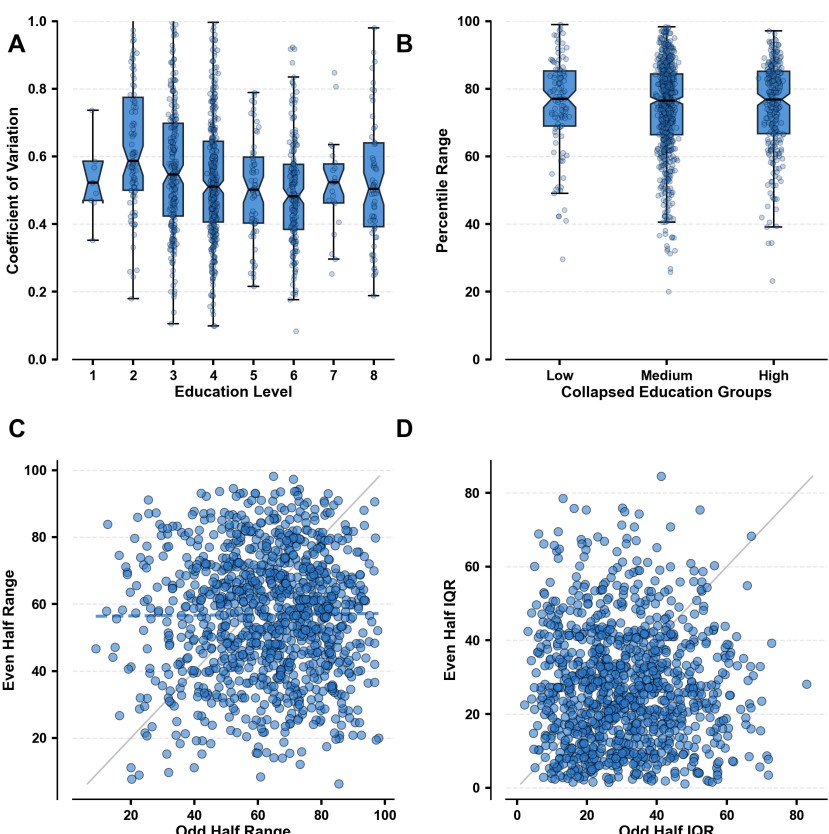

Figure 4: **Sensitivity analyses confirm the absence of educational effects on cognitive profile heterogeneity across multiple analytical approaches and metric definitions.** Alternative heterogeneity metrics and analytical strategies yielded consistent null findings, reinforcing the robustness of the primary results. The coefficient of variation metric (Panel A) showed no systematic relationship with educational attainment, mirroring patterns observed with the primary percentile-based measures. Collapsed education groupings (Panel B) revealed overlapping distributions with minimal between-group differences, further supporting the absence of education-related cognitive differentiation effects. Split-half reliability analyses (Panels C and D) demonstrated adequate internal consistency of heterogeneity metrics. Panel A displays coefficient of variation values across eight education levels, with individual data points (blue circles) overlaid on box plots. Panel B shows percentile range distributions for three collapsed education groups (Low, Medium, High), with individual participants represented as semi-transparent blue points. Panels C and D present split-half reliability assessments, plotting odd-numbered subtests against even-numbered subtests for range and IQR metrics respectively. Gray diagonal lines represent perfect reliability (r = 1.0), while dashed regression lines show observed correlations with coefficients displayed in legends. Coefficient of variation calculated as $\sigma/\mu$ of each participant's 11 age-stratified percentile scores. Education groups collapsed as: Low (levels 1-2), Medium (levels 3-4, 8), High (levels 5-7). Split-half correlations computed using Pearson's r between odd-numbered subtests (36, 40, 28, 30, 32, 37) and even-numbered subtests (39, 29, 33, 27, 38). All analyses applied identical exclusion criteria and statistical assumptions as primary models. Sample size n = 1,083 participants across all panels.

Table 2: **Education-age interaction effects on cognitive profile heterogeneity show no significant cumulative differentiation across lifespan groups** Multiple regression analysis examining whether educational attainment effects on cognitive heterogeneity (percentile range and interquartile range of performance across 11 cognitive subtests) differ between age groups (Younger 18-39 years, n=530; Middle 40-49 years, n=170; Older 50+ years, n=383) with education collapsed into Low (some high school/high school diploma), Medium (some college/college degree/associate's degree), and High (professional degree/master's degree/Ph.D.) categories, using Low education and Younger age as reference groups (N=1,083 total). Standardized beta coefficients shown with standard errors in parentheses, 95% confidence intervals in brackets; no interaction terms reached significance after Bonferroni correction ($\alpha = 0.025$), indicating that education-heterogeneity relationships remain consistent across the adult lifespan rather than showing predicted cumulative differentiation effects in older adults.

| Model/Predictor | Std. Beta (SE) | 95% CI | p-value |
|---|---|---|---|
| **Model 1 Range: Percentile Range** | | | |
| *Main Effects* | | | |
| Education: High | -0.018 (2.266) | [-5.012, 3.881] | 0.803 |
| Education: Medium | 0.019 (1.942) | [-3.264, 4.356] | 0.778 |
| Age: Middle | 0.026 (4.006) | [-6.879, 8.841] | 0.807 |
| Age: Older | 0.136 (3.070) | [-2.147, 9.900] | 0.207 |
| *Interaction Effects* | | | |
| Education: High × Age: Middle | 0.027 (4.598) | [-7.400, 10.644] | 0.724 |
| Education: Medium × Age: Middle | -0.037 (4.282) | [-10.189, 6.617] | 0.677 |
| Education: High × Age: Older | -0.052 (3.578) | [-9.329, 4.714] | 0.519 |
| Education: Medium × Age: Older | -0.132 (3.255) | [-10.742, 2.031] | 0.181 |
| *Model Fit Statistics* | | | |
| $R^2$ | 0.013 | | |
| Adjusted $R^2$ | -0.001 | | |
| F-statistic | 0.929 | | 0.531 |
| N | 1083 | | |
| **Model 2 IQR: Percentile IQR** | | | |
| *Main Effects* | | | |
| Education: High | -0.008 (1.918) | [-3.967, 3.560] | 0.915 |
| Education: Medium | -0.014 (1.643) | [-3.566, 2.883] | 0.835 |
| Age: Middle | 0.078 (3.390) | [-4.180, 9.126] | 0.466 |
| Age: Older | -0.014 (2.598) | [-5.431, 4.766] | 0.898 |
| *Interaction Effects* | | | |
| Education: High × Age: Middle | -0.020 (3.892) | [-8.675, 6.597] | 0.789 |
| Education: Medium × Age: Middle | -0.030 (3.625) | [-8.339, 5.885] | 0.735 |
| Education: High × Age: Older | -0.026 (3.029) | [-6.934, 4.952] | 0.744 |
| Education: Medium × Age: Older | -0.000 (2.755) | [-5.419, 5.392] | 0.996 |
| *Model Fit Statistics* | | | |
| $R^2$ | 0.019 | | |
| Adjusted $R^2$ | 0.005 | | |
| F-statistic | 1.386 | | 0.146 |
| N | 1083 | | |


