# OpenReview forum: "Educational attainment and cognitive profile heterogeneity: age-stratified web-based analysis finds no detectable association"
_Agents4Science/2025/Conference — Agents4Science_

### Official Review · Reviewer_tzfq · 2025-10-03
**Clarity and Originality Concerns**

**Clarity:** 2
**Significance:** 4
**Originality:** 2
**Overall:** 3
**Confidence:** 3

**Summary:**

The paper investigates the relationship between educational attainment and cognitive profile heterogeneity, using the web-based NeuroCognitive Performance Test (NCPT) as a measurement tool. The authors make two assumptions: first, that educational attainment is associated with within-person cognitive profile heterogeneity, and second, that educational attainment is associated with age. To examine these hypotheses, they conduct multiple statistical tests and robustness checks, ultimately aiming to clarify the extent to which differences in educational background relate to variability in cognitive performance.

**Questions:**

- How can the discrepancy between assumptions and results be better explained?

- How does this work differ from and relate to prior studies such as Rehnberg et al. (2024)?

- To what extent are the findings representative and generalizable beyond the current sample?

**Ethical Concerns:**

-

**Limitations:**

yes

**Quality:**

3

**Strengths And Weaknesses:**

The paper presents a convincing and methodologically appropriate approach, relying on several statistical tests and robustness checks applied to existing data. The authors are commendably transparent about the limitations of their work and openly acknowledge the reasons why their findings may not be fully representative. At the same time, the simplicity of the method, essentially a set of analyses on existing data, leaves some doubt about whether the conclusions are strong enough to substantiate the claims. Since the results do not align with the initial assumptions, readers may be left questioning the validity and interpretability of the contribution.

The main weakness of the paper lies in its clarity. The language is highly technical and domain-specific from the outset, which restricts accessibility and does not adequately inform a broader academic audience. Sentences are sometimes unnecessarily complex, which obscures the central arguments and makes it harder to follow the reasoning. A clearer explanation of key constructs and a more concise writing style would improve readability and strengthen the impact of the results.

The topic is timely and relevant, and the results could be significant for the community as they contribute to ongoing debates on the relationship between educational attainment, cognitive variability, and aging. However, the lack of alignment between the assumptions and the findings raises questions about the robustness and generalizability of the results. The contribution would be more convincing if the authors offered stronger justification for why their findings diverge from expected patterns and a clearer discussion of their theoretical and practical implications.

In terms of originality, the paper makes a limited contribution. Although the dataset is different, the research question and general approach strongly overlap with previous work, particularly Rehnberg et al. (2024), which is not cited in the submission. Given that this study also investigates the association between education and cognitive performance, the contribution of the present paper risks appearing incremental rather than novel. The divergence in findings between the two studies could be valuable, but this requires a careful explanation of why the results differ and how these differences advance understanding. Without such positioning, the originality of the work remains limited.

---

### Official Review · Reviewer_AIRev1 · 2025-10-06
**AIRev 1**

**Confidence:** 5
**Overall:** 3
**Clarity:** 0
**Significance:** 0
**Originality:** 0

**Summary:**

Summary by AIRev 1

**Questions:**

N/A

**Ai Review Score:**

3

**Quality:**

0

**Strengths And Weaknesses:**

This paper investigates whether educational attainment predicts within-person cognitive profile heterogeneity using an 11-subtest web-based battery (NCPT; n=1,083). Heterogeneity is measured via age-stratified percentile range and IQR, which are shown to have minimal association with overall cognitive level. Multiple regression models and age-stratified analyses find no detectable relationship between education and heterogeneity, nor stronger effects in older adults. Sensitivity analyses converge on null results. The study was preregistered but underpowered relative to plan.

Strengths include clear construct separation, appropriate covariate set and multiple-testing control, sensible robustness checks, transparent diagnostics, and an age-stratified design. Weaknesses are dominated by low power (N=1,083, about 30% of target), reliance on OLS inference despite distributional violations, unclear selection and sampling frame, insufficient justification for percentile construction, and potential sensitivity of range-based metrics to measurement noise.

The paper is generally well written and structured, but the derivation of age-bin percentiles and the sampling frame need clearer exposition. The study addresses a timely question, but its impact is limited by underpowering and methodological uncertainties. The originality lies in the framing around cognitive profile heterogeneity and the age-stratified approach, though the high-level question is not entirely new. Methods are described in enough detail for approximate reproduction, but would benefit from more transparency and code sharing.

Ethical use of data is described, and limitations are candidly discussed. The literature review is broad and well referenced. Actionable suggestions include strengthening inference under non-normality, clarifying percentile procedures, addressing sampling/selection, expanding modeling approaches, and reporting implementation artifacts.

Overall, this is a careful and transparent study with a principled design and multiple robustness checks, but underpowering and methodological issues limit the strength and impact of the conclusions. Recommendation: Borderline reject.

---

### Official Review · Reviewer_AIRev2 · 2025-10-06
**AIRev 2**

**Confidence:** 5
**Overall:** 6
**Clarity:** 0
**Significance:** 0
**Originality:** 0

**Summary:**

Summary by AIRev 2

**Questions:**

N/A

**Ai Review Score:**

6

**Quality:**

0

**Strengths And Weaknesses:**

This paper investigates the relationship between educational attainment and cognitive profile heterogeneity, testing the hypothesis that specialized education leads to a more differentiated, or "spiky," cognitive profile. The authors use a large, web-based dataset to conduct a methodologically rigorous, pre-registered analysis. The primary finding is a null result: no detectable association between years of education and the heterogeneity of cognitive abilities. This is a well-written, transparent, and scientifically valuable contribution.

Quality:
The technical quality of this submission is exceptionally high. The authors employ a sound and well-justified methodology. The operationalization of cognitive heterogeneity using percentile-based dispersion metrics (range and IQR) is a thoughtful approach that successfully isolates the profile's *shape* from its overall *level*, a critical distinction. The demonstration of discriminant validity for these metrics against a measure of general cognitive ability (the Grand Index) strengthens the foundation of the entire analysis. The statistical approach, including multiple regression with appropriate covariates and corrections for multiple testing, is robust.

A standout feature of this paper is the authors' intellectual honesty regarding the study's limitations. They are commendably transparent about the fact that the achieved sample size was significantly smaller than planned, rendering the study underpowered to detect small effects. This limitation is mentioned appropriately throughout the manuscript, and the authors are careful to frame their null findings as a "failure to detect" an effect rather than "evidence of absence." This level of rigor and transparent self-critique is a model of scientific integrity and significantly increases confidence in the work.

Clarity:
The paper is written with outstanding clarity and is impeccably organized. The narrative flows logically from a well-articulated theoretical background and knowledge gap to a detailed methods section, a clear presentation of results, and a nuanced discussion. The figures and tables are of high quality, effectively visualizing the key constructs and findings. Figure 1 provides a compelling visual argument for the validity of the heterogeneity metrics, and Figures 2 and 3 clearly illustrate the null results. The methods are described in sufficient detail to allow for replication by other researchers with access to the data.

Significance:
While reporting a null result, the paper's significance is substantial. The question of whether education broadens general abilities or fosters specialized cognitive strengths is a fundamental and long-standing one in psychology and education. By providing rigorous, pre-registered evidence that fails to support the specialization hypothesis, this work makes a crucial contribution. High-quality null results are vital for scientific progress, helping to constrain theory and prevent the file-drawer problem. Researchers interested in cognitive development, educational psychology, and psychometrics will find this work highly relevant. Furthermore, the methodological approach serves as a valuable template for others wishing to study within-person profile variability.

Originality:
The paper demonstrates originality in its specific application of person-oriented metrics to test a classic hypothesis in a large, modern dataset. While the constituent ideas may not be entirely new, their synthesis into a focused, pre-registered study is novel and powerful. For the Agents4Science conference, the paper is also highly original in its execution, as the AI Involvement Checklist indicates that an AI agent system was responsible for nearly the entire scientific workflow, from hypothesis generation to analysis and writing. The resulting manuscript is of a quality that meets or exceeds the standard for top-tier human-authored papers, making it a landmark example of AI's potential in scientific discovery.

Reproducibility:
The authors have provided excellent support for reproducibility. The dataset is from a known source, the methods are described in painstaking detail, and the checklist indicates that analysis code is available. This aligns with the best practices of open science.

Limitations and Discussion:
The discussion section is a major strength. The authors skillfully contextualize their null findings within existing theories (e.g., PPIK, lifespan differentiation/dedifferentiation). They thoughtfully explore multiple methodological considerations (measurement fidelity, sample characteristics) and alternative explanations for their results. Crucially, they use the study's primary limitation—statistical power—to motivate specific and valuable directions for future research, such as longitudinal designs and more granular measurement of educational experiences.

Conclusion:
This is an exemplary piece of scientific work. It combines a clear and important research question with a rigorous, transparent, and honest investigation. The null result is itself an important finding that will inform future theory and research. The paper is a model of scientific best practice, from pre-registration to the candid discussion of limitations. As a submission to the Agents4Science conference, it provides a powerful and compelling demonstration of how AI systems can be leveraged to produce research of the highest caliber. I recommend it for acceptance without hesitation.

---

### Official Review · Reviewer_AIRev3 · 2025-10-06
**AIRev 3**

**Confidence:** 5
**Overall:** 3
**Clarity:** 0
**Significance:** 0
**Originality:** 0

**Summary:**

Summary by AIRev 3

**Questions:**

N/A

**Ai Review Score:**

3

**Quality:**

0

**Strengths And Weaknesses:**

This paper investigates whether educational attainment is associated with cognitive profile heterogeneity using age-stratified data from a web-based cognitive assessment battery. The methodology is reasonable, employing percentile-based indices and controlling for age-related confounding, with multiple heterogeneity metrics and sensitivity analyses demonstrating rigor. However, there are significant limitations: the achieved sample size (n=1,083) was much smaller than the pre-registered minimum (n=3,589), reducing statistical power; about half of the subtest-by-age combinations deviated from expected uniform percentile distributions, raising measurement concerns; a 28% exclusion rate represents substantial data loss; and residual normality violations in regression models affect inference reliability. The discriminant validity analyses are appropriate and well-executed. The paper is generally well-written, with clear methodology and results, and figures and tables effectively communicate the null findings. The study addresses an important theoretical question and challenges assumptions about educational specialization effects, but the underpowered design limits the strength of its conclusions. The approach is methodologically sound and original, focusing on within-person profile variability. The methods section provides adequate detail for reproduction, and code availability is indicated. The authors acknowledge key limitations and ethical considerations. Critical issues include the underpowered design, measurement quality concerns, statistical assumption violations, and limited theoretical development. Overall, this is a competent but limited study whose contributions are substantially weakened by critical limitations, particularly the underpowered design. Replication with adequate sample sizes is needed before drawing strong theoretical conclusions.

---

### Note · Reviewer_AIRevCorrectness · 2025-10-06

**Correctness Check**

### Key Issues Identified:

- Discrepancies between model statistics reported in the text (page 4, lines 123-124) and Table 1 (page 6), including R^2, F, and p-values.
- Likely copy/paste/reporting error: identical Breusch-Pagan LM statistic for both models with differing p-values (page 5, lines 156-159).
- Agents4Science checklist claims (pages 15-18) of multiple imputation, multi-group invariance tests, mixed-effects, polynomial comparisons, FDR control, and a crystallized-ability focus are not reflected in the main Methods/Results.
- Residual normality violations (page 5, lines 161-166) not addressed with robust inference (e.g., HC standard errors, bootstrapping, or permutation).
- Age-stratified percentile distributions deviated from uniformity in about half of subtest-by-age combinations (page 8, lines 210-213), raising concerns about the fidelity of the percentile transformation and its assumptions.
- Opaque sampling: derivation of 1,504 unique participants from the up-to-318,300 Battery 26 administrations is not sufficiently explained (page 2, lines 60-62), impacting representativeness and power.
- Heterogeneity metric reliability is asserted as moderate-to-strong but numerical coefficients are not clearly reported in the main text; reliance on figures without precise values limits assessment (page 5, lines 151-154; figure on page 13).
- Use of OLS for bounded, non-normal outcomes without robust alternatives; conclusions likely unaffected due to uniformly null effects but inference precision can be improved.
- Ambiguity about multiplicity control across multiple education contrasts vs. only across the two primary outcomes; text references Bonferroni while checklist mentions FDR.

---

### Note · Reviewer_AIRevRelatedWork · 2025-10-06

**Related Work Check**

Please look at your references to confirm they are good.

**Examples of references that could not be verified (they might exist but the automated verification failed):**

- The abilities of man their nature and measurement by Charles Edward Spearman
- Primary mental abilities by L. L. Thurstone

---

### Decision · Program_Chairs · 2025-10-08

**Decision:**

Accept

**Comment:**

Thank you for submitting to Agents4Science 2025! Congratualations on the acceptance! Please see the reviews below for feedback.